# Adaptation Process of Korean Fathers within Multicultural Families in Korea

**DOI:** 10.3390/ijerph18115935

**Published:** 2021-05-31

**Authors:** So-Yeon Park, Suhyun Kim, Hyang-In Cho Chung

**Affiliations:** 1Department of Nursing, Chunnam Techno University, Jeollanamdo 57500, Korea; mosuamam@cntu.ac.kr; 2Department of Nursing, Nambu University, Gwangju 62271, Korea; ksh136112@nambu.ac.kr; 3College of Nursing, Chonnam National University, Gwangju 61186, Korea

**Keywords:** fathers, grounded theory, Korea, multicultural families, psychological adaptation, qualitative research

## Abstract

This study aimed to develop a grounded theory regarding the associations between factors identified in the adaptation process of 11 fathers of multicultural families. Participants were recruited purposively and data were collected through in-depth interviews. Data analysis was performed alongside data collection, following Strauss and Corbin’s grounded theory method. The following 11 categories were derived from the data: (1) a reluctantly formed multicultural family, (2) finding happiness amid confusion, (3) cultural differences, (4) economic difficulties, (5) social prejudice and alienation, (6) restrictions on the use of local services, (7) facing multiple obstacles, (8) people who provided strength and support, (9) accepting differences and moving forward, (10) growth with family, and (11) being made to stay. The core category was identified as “struggling to protect the family with a double burden.” This theory described the adaptation process of being a father in a multicultural family and participants’ reactions to the circumstances around it. Based on this theory, health policies should be developed to support not only the expansion of medical facilities in island and rural areas but also the activation of self-help groups. Future research should include the development of web-based prenatal management and parental education in immigrants’ native languages.

## 1. Introduction

With advances in technology, globalization is rapidly progressing, marked by cross-border movement and activities worldwide. Even in Korea, an ethnically homogeneous nation, foreigners began to migrate to the country in earnest in the 1990s. Currently, Korea is a multicultural country as the number of foreigners staying in Korea exceeded 2.3 million in 2020 [1]. Most of the foreigners who live in Korea are married migrant women, and as a result, families are becoming more diverse due to cultural diversity and changes in the family structure [2]. In particular, the number of multicultural families created through marriage with migrant women in Korea increased from 44,000 in 2007 to 122,000 in 2010 and 237,000 in 2018 [3]. Accordingly, the government is redefining the term “healthy family” and making efforts to improve laws and systems in accordance with various family types and changes in the family environment [2]. Similarly, in the nursing field as well, it is necessary to expand and approach multicultural families as new recipients of nursing care so that they too can take root as healthy families within the Korean society.

Multicultural families formed by migrant women in Korea have different characteristics from similar families in other countries. To create a healthy multicultural family, the husband’s role is very important [4,5], because it affects not only the relationship with the migrant spouse [4] but also the cognitive, emotional, and social development of their children [5]. Thus, fathers in multicultural families are fosterers, cultural negotiators, and cultural links within their families, supporting married migrant women who are struggling with cultural adaptation and child rearing in Korean society, and have a direct impact on child development [6,7].

However, in the process of becoming fathers, they are struggling not only with conflicts with spouses who have grown up in different environments with different cultural backgrounds, but also with social perceptions of minorities in Korean society, which takes pride in being an ethnically homogeneous nation [6,7]. Korean men who marry migrant women at an older age experience stress due to the lack of preparation for marriage, the generational and age gaps with foreign wives, and an unstable marital life, which is reflective of the nature of brokered marriages [6]. Korean men who pay a broker and marry a foreign woman are generally of an older age, and many live locally or in rural areas. They are often not involved in raising their children because they are highly aware of and espouse the traditional role of fathers in Korea. In addition, they cannot actively participate in parenting because their income is low, and they have to devote a lot of time to work to maintain economic stability [7]. 

In general, the period in which family members experience the greatest role changes in the family life cycle are during the birth and rearing period of a child. There is greater change in this period than in any other stage of development [8,9,10]. Men experience parenthood through the birth of their children, and they feel the need for active participation and experience enormous responsibility in assuming the dual role of a husband and father [6,7]. If men cope with these times incorrectly, the relationship between parents and children may be impaired or the children may develop social and mental problems that can have a negative impact on their growth and development [9]. At such an important time, fathers of multicultural families are required to play more roles and bear more responsibilities than fathers of homogenous families [8,9]. For maintaining the social health of multicultural families adequately, it is necessary to pay attention to the various problems they experience. Above all, the process of having their first child and becoming a father in a multicultural family is inevitably a great challenge for them [9]. Therefore, a family nursing approach that reflects the context of multicultural families based on an understanding of the process of paternity adaptation of fathers of multicultural families is needed, and a substantive theory developed by grasping the nature of this relationship as it interacts with related conditions.

Previous studies of multicultural families during the birth and rearing period have mainly considered and investigated the process of maternal adaptation [11,12,13], and parental education programs have mostly targeted foreign mothers [14]. Research on fathers has been limited to those belonging to ethnically homogenous families [15,16,17], and quantitative studies have been conducted focusing on variables related to fathers’ parenting stress and role performance in multicultural families [18,19,20]. However, it is difficult to generalize the process of adaptation to the father’s role in multicultural families because the contextual situation is different from that of ordinary families, and quantitative studies promoting our understanding of their experiences as a whole are limited. In particular, in the nursing field, which plays the role of primary health care in the community, it is necessary to understand the process of adaptation to fatherhood to improve the health care of multicultural families in local communities. In other words, research is needed to closely examine fathers’ experiences, attain a practical and comprehensive understanding of the same and thoroughly grasp and confirm various conditions that interact in the process of adaptation to fatherhood in such families.

The purpose of this study was to understand the process whereby fathers of multicultural families adapt to fatherhood during the infancy of their first child, and to explore how various conditions affect the adaptation process of such fathers to derive a substantive theory.

## 2. Materials and Methods

### 2.1. Study Design

This is a qualitative study that applied Corbin and Strauss’ grounded theory method [21] to understand the paternity adaptation process and related conditions from the perspective of Korean fathers in multicultural families.

### 2.2. Participants

The inclusion criteria of participants in this study are as follows: a Korean man (1) who had married a foreign woman through an intermediary, (2) whose first child was under 3 years old. Participants in the study were recruited using a theoretical sampling method until a state of saturation was reached where no new concept emerged. In total, 11 participants were selected. Fathers with children below 3 years were selected because this is the period wherein parents have the greatest influence on the physical and emotional development of the child [4], and it is early in this period of the adaptation process that men become fathers [4,5].

Study participants’ ages ranged from 43 to 59 years, with an average age of 48 years, and their spouses’ ages ranged from 22 to 40 years, with an average age of 31 years. Participants’ educational backgrounds ranged from elementary school dropouts to graduates of junior college; they were engaged in various occupations such as construction work, cargo transportation, and fixed-term public servants. Six people lived in the downtown area and five people lived in outlying areas, and all of them had nuclear families. Children’s ages varied from 4 months to 31 months after birth (Table 1).

### 2.3. Recruitment

Data were collected through in-depth one-on-one interviews with the research participants, from 13 July to 31 August 2017. The first author who had professional training in qualitative research method was in charge of collecting the data. For this purpose, the first author visited the Multicultural Family Support Center, explained the purpose of the research and the procedure for collecting data to the head of each institution, and asked to be introduced to potential participants. Research participants were chosen through the snowball sampling method. 

Researchers interviewed participants in their homes, multicultural centers, and health centers, as per their preference. The interview time and place were decided based on the participants’ convenience.

The interview questions were semi-structured and open-ended. To establish a relationship of trust with the participants and create a comfortable atmosphere, the interview proceedings began with general conversation and gradually led to the subject of the study. The first question on the subject was open-ended; an example would be, “What was the process of becoming a father like, after your first baby was born?” The researcher allowed participants to talk about their experiences as much as possible and gradually asked specific questions as needed. Each interview took about 1 hour to 1 hour and 30 minutes, and participants were interviewed 1 to 2 times each. In the process of data collection and analysis, additional interviews were conducted when there were concepts requiring further confirmation from the research participants. Overall, there were 5 participants (45%) with whom more than 2 interviews were conducted. Interview data were recorded using a mobile phone or a recorder with the consent of participants and terminated when it was judged that theoretical saturation was reached, i.e., if no new categories appeared during the data analysis process. In addition to the interview and transcription data of the participants, data were collected through the researcher’s field notes and memos.

### 2.4. Data Analysis

Data analysis was conducted alongside data collection, and open coding, axis coding, and selective coding were performed step by step according to the grounded theory method suggested by Corbin and Strauss [21]. After the interview, a researcher tried to complete the transcription work within 24 hours. However, since the study participants lived in the island area, transportation time was limited; thus, the interview transcriptions of a few study participants took up to 48 hours. By repetitively reading the transcribed interviews, the field notes created during the interviews, and the memos, the data were analyzed line by line, concepts were classified, and similar concepts were organized into subcategories. In addition, the relationships between categories derived through open coding were organized into a paradigm model that includes causality, situationality, phenomenon, intervening conditions, strategies, and outcomes based on the dimension of categories and attributes. By thinking about the relationship between the concepts and subcategories obtained through the analysis, the core category was derived and propositions were derived through relational statements, thus leading to a substantial theory that could explain the paternal adaptation process of Korean fathers in multicultural families.

### 2.5. Study Validity and Researcher Preparation

This study secured validity by applying the four criteria of reliability, applicability, consistency, and neutrality as suggested by Guba and Lincoln [22]. First, reliability was secured by confirming whether the research result accurately measured the reality. The researcher rechecked the transcribed interviews multiple times so that no interview data were omitted. In addition, by continuously comparing the transcribed data with the results of the analysis process, efforts were made to secure reliability by repeatedly checking whether the classifications into concepts, etc., were consistent. In particular, the first author has been working for 30 years in a women’s hospital in a small city and has gained expertise and sensitivity in terms of the study subject. Data were collected by the first author who had undertaken professional training in qualitative research. In the process of data collection, with sensitivity to the context and meaning, the first author corrected and supplemented the information on the adaptation process provided by the participants by consulting with five nursing professors with extensive experience in qualitative research. Second, since the applicability measures the possibility of generalization of the research results, various participants were deliberately selected and new concepts that appeared in the interview process were confirmed. In addition, we tried to reflect the similarities and differences of the data in-depth, by collecting data until saturation. The research results were also compared with data from two fathers belonging to multicultural families who did not participate in the research. Third, consistency was ensured by describing the process of analysis in detail during data collection and striving to strictly observe the procedure. Five nursing professors with extensive experience in qualitative research were asked to evaluate the analysis process. Fourth, as neutrality means objectivity for the research process and results, a literature review was conducted to a certain extent after the data collection and analysis so that the contents of the literature review would not lead the researcher to prejudge the findings or prejudice him toward certain conclusions. In addition, to reflect the participants’ experiences as accurately as possible and to prevent the researcher’s past experience from influencing the study, the researcher confirmed with the participants what was understood and verified whether participants’ perceptions of the adaptation process matched the researcher’s analysis.

### 2.6. Ethical Consideration

Research approval was obtained from the University Bioethics Review Committee (IRB NO: 1040198-170519-HR-043-02). Before beginning the study, the researcher explained the purpose and method of the study to the participants and obtained informed consent. Before obtaining written informed consent, the researcher explained that the participant could end the interview process at any time, and the time required for the interview was also mentioned in advance. The researcher also explained to the participants that the interviews would be recorded, the recorded and transcribed data would be used only for research purposes, and the anonymity of the research participants was guaranteed.

## 3. Results

Categorization of the interview data through an open coding process yielded 30 subcategories, and 11 higher-level categories (Table 2). The relations of categories were established through axis coding according to the paradigm model, and a context model was also presented by combining all categories (Figure 1). In this study, the paternal adaptation process of the fathers of multicultural families was found to comprise “confusion”, “acceptance of reality”, “adaptation to reality”, and “growth”, and the core category was “struggling to protect the family with a double burden”.

### 3.1. Causal Conditions

The causal condition of Korean fathers’ paternal adaptation process was found to be “multicultural family reluctantly formed”, whose subcategories were “continuous suggestions from family and neighbors”, “unwilling marriage”, and “being a father by compelling circumstances”. The study participants expressed reluctance to marry foreign women under the continuous suggestions of family and neighbors so as to accomplish the developmental tasks required at that point in their life cycle. This made it burdensome to even have a plan for children, but they were compelled by the situation to become fathers.


*Wouldn’t I also want to marry a Korean woman? I gave up because I had no money and I was getting older. So, I couldn’t marry a Korean woman, so I thought I had to pay the brokerage and marry a foreign woman from a developing country.*
*(Participant 3)*


*When I was introduced by a marriage company, the biggest reason I was forced to get married was my age; I was over 50. When I turn 60, my child will be 10, but when the child becomes an adult, it seems that there will be a lot of pressure on the child, so I tried not to have a baby. But when I had a baby, I was forced to accept it.*
*(Participant 8)*

### 3.2. Central Phenomenon 

In this study, the central phenomenon that fathers from multicultural families experience in the process of paternity adaptation was “finding happiness amid confusion”. The subcategories for this were “the confusion of being a father” and “the thrill of being a father”. The former subcategory involved not only the implications of the role of a father, but also the anxiety of how to raise the child and a sense of responsibility for the family. Participants felt a combination of fear and difficulty in parenting activities due to their lack of parenting skills, a sense of great responsibility as the head of the household after childbirth, and regret for not being able to be with the child due to poverty and the need to work. Most participants experienced the role of father without being habituated to the role of husband and prepared for parenthood through the developmental stages of newlyweds and nurturing that led to pregnancy and childbirth immediately after marriage. A father with a foreign spouse has a dual responsibility as a husband who must not only adapt to being a father but also help his wife adjust to life in Korea as she experiences linguistic and cultural differences. Another aspect discovered was “the thrill of being a father”, at being responsible for a child. 


*I liked having a child, but my heart was torn. I am sorry that the bigger my child gets, the more responsibility I have to feed the child. I wanted to spend time with my child, but it was too busy and difficult because I had to make money. So, I couldn’t do that.*
*(Participant 6)*


*Even though I wanted to quit because the work was so hard, I remembered when my child was just born, and I felt strong. As soon as the baby was born, the baby went straight into the newborn room, and the first time I saw the baby it was very small. The nurse told me to hug the baby, but my heart was so overwhelmed and I was tearing up, so I couldn’t hold him properly.*
*(Participant 1)*

### 3.3. Contextual Conditions

In this study, the contextual conditions that influence the process of paternity adaptation were “cultural differences”, “economic difficulties”, “social prejudice and alienation”, and “restrictions on the use of local services”. Participants found it difficult in terms of not only language barriers as cultural differences but also economic circumstances and child rearing. Even if the medical institution from which they received prenatal care provided education on child rearing, it was incomprehensible to the spouse due to the language barrier, resulting in her lack of parenting information and skills. In addition, parenting stress was high because couples could not reach a consensus regarding raising their children due to language barriers.


*Too often, my wife doesn’t know anything about parenting. She simply skipped the regular schedule of immunizations for babies. Even if the hospital sent home an immunization mail, my wife couldn’t read the text. If my wife were a Korean woman, she could call around and ask. But my wife is a foreigner, so she doesn’t know.*
*(Participant 5)*


*When my wife was pregnant, I was uncomfortable because I couldn’t understand what was wrong. We live on an island, so there are no hospitals. I asked my wife with gestures and talked. At first, I thought I couldn’t even do this anymore.*
*(Participant 2)*


*When the baby was born and less than one year old, it was sick from a cold. But my wife applied tiger medicine to the baby’s head over the pores. I fought with her a lot at that time. She said that in her country they do that. My wife also applied the medicine to the baby’s belly. These days, tiger medicine is not used in Korea. But still, in my wife’s country, they apply it on the head of a 5- or 6-month-old baby and also on the belly.*
*(Participant 4)*


*I said that my home situation is difficult economically, so I have to save. But my wife said she didn’t save money in her country, so she didn’t understand. When I think about it, it seems that in my wife’s country, which is a developing country, she was too poor to be able to save.*
*(Participant 4)*

In terms of “economic difficulties”, as the study participants aged, supporting both their own and their wife’s families was burdensome. The average age of the participants was 48, and they were anxious that they had relatively fewer work years left. In addition, they had to bear the cost of raising a child. Some participants tried to get help in raising their children by inviting their spouse’s family or mother from abroad, if they did not have the support of their own Korean family. However, this did not solve their problems of raising their children, and as the spouse’s family also needed support, this increased their economic burden. In such economic crisis, providing financial support to the wife’s parents was an additional psychological burden.


*The bigger the baby gets, the more I am worried about taking care of them. If I buy clothes for my baby, I still have to have clothes in the future, and though I want to dress well with pretty and nice clothes, I won’t have the money. As the baby grows, it will continue to cost a lot of money. But I haven’t saved any money. When I think about it, my heart is upset. Sometimes it’s comfortable not to think about it, so I just live like that.*
*(Participant 2)*


*Anyway, I can’t afford it since I earn the money alone. Because I make money driving a taxi, there is no money to be saved, and I have to spend all that I make.*
*(Participant 6)*


*I brought my wife’s mother from abroad to Korea, hoping that she would take good care of the baby. But my wife’s mother was sick and my money was spent for hospital bills.*
*(Participant 11)*

Social prejudice toward multicultural families and societal alienation also appeared to be contextual conditions influencing the process of becoming a father. The participants worried that their children would be bullied for their different skin color and appearance. Some participants were concerned about the negative perceptions held by Korean society regarding multicultural families and even experienced uncomfortable gazes from others. Participants compared the children born within ethnically homogenous Korean families with their own children and thought that their wives had caused bilingual confusion for the children when their language development was late. They expressed anxiety and concern about not only their children’s different appearance but also societal perceptions of their children. 


*Since I am marrying a foreigner rather than a Korean, the biggest issue I think is that my child looks like a foreign child. As children grow older, their skin color and appearance is slightly different from normal [ethnically, homogenously Korean] children. So, as they grow older, I am worried that they will be alienated from their peers.*
*(Participant 5)*


*I know my friends hate talking about my marriage to a foreign wife. So, they don’t directly express that my child is different, but they say this indirectly. “Your baby seems like a foreigner.” I’m upset by those words. So, I don’t meet more friends.*
*(Participant 6)*


*One of the reasons I hesitated to marry internationally is because of how others look at it. I feel like people point at me, saying, “He bought a wife with money,” and look at me badly. When I went to a restaurant with my wife and she spoke to her mother in a foreign language, it seemed like everyone was looking at me. Multiculturalism itself is a stranger here.*
*(Participant 10)*

The subcategories of “restrictions on the use of local services” that appeared as contextual conditions were “medical services in islands with low accessibility” and “services from community organizations that are not easy to use.” Because the study participants lived on islands, obstetricians and gynecologists were not easily accessible for prenatal examinations. Although the government provides a postpartum helper service as a childbirth support policy, it was difficult to take care of women from multicultural families because the actual postpartum helper could not come to the islands where participants lived because of the difficulty in getting there. Some participants even took a boat to the mainland on weekends to receive education on raising children. However, there was no specialized and diverse education for parenting in their region, so the burden and stress of parenting intensified.


*Since we live on the island, there is no hospital that we can use quickly even if our [my] wife and child are sick. There are no health clinics on the island where we live. People around me tell me to go to a pediatric hospital because they are careful about babies, but there is no hospital. It’s difficult because I have to go to the hospital with my wife and child. I have to go to work. My wife doesn’t speak [Korean] well, so I can’t tell her to go alone.*
*(Participant 4)*


*I want to receive education about parenting together for my child on the weekend, but the educational institution is closed on the weekends. I don’t want to do that every day, but once a month, I wish I could use the weekend for education. However, since all education is provided during the day on weekdays, and I have to go to work, this is not possible. This is a problem.*
*(Participant 11)*


*My wife is alone and is taking care of the baby, so I hope someone can come and help me if something goes wrong. Of course, it would be hard to be with my wife who doesn’t speak the language, but I wish I had someone who could help me when my baby is sick or needs help from time to time, not every day. But who can get to the island?*
*(Participant 1)*

### 3.4. Intervention Conditions

This study found that fathers of multicultural families faced several obstacles, classified as intervention conditions, which aggravate problems in the process of paternity adjustment. The subcategories for this were “difficulties in raising children” and “conflict within families.” Participants faced difficulties in raising children without receiving education on the subject. They received great help from their families, especially from their mothers, in the early stages of parenting. They experienced conflict with their mothers, who were dissatisfied with a daughter-in-law from a different culture who spoke a different language and were stressed by it.


*In the first few months after the baby was born, I couldn’t hold my baby because it felt like a threat to him. It’s not that I hate babies. I was afraid because I didn’t know how to hold the baby and had never done it before.*
*(Participant 8)*


*After the baby was born, I kept talking to the baby in Korean and my wife did it in Vietnamese. I was worried that it would cause confusion in my baby’s language development.*
*(Participant 2)*


*My child’s language development was a little late. Still, the child said “Dad” and “Mom.” About 30 months after birth, our child was about to start talking, but my wife went to Cambodia with him. Since then, the child has been confused and he is still unable to speak. I was so worried that I went to the speech therapy center. The speech therapist consulted with my child for 30 minutes and told me to pay 100,000 won. Due to the burden of the cost, I gave up receiving such speech therapy. When I tried to ask for speech therapy from the government or province, the government didn’t give it to us.*
*(Participant 3)*


*My wife often fought with my mother. My wife is from a foreign country, so she doesn’t know Korean, but I think she had a different idea about raising a baby than my mother. I had a hard time every time these two quarreled. I understood my mother’s heart as well as my young wife’s heart.*
*(Participant 5)*

“People who provided strength and support” was an intervening condition that alleviated the problems experienced by the fathers in the process of paternity adaptation. The subcategories for this were “the presence of a child”, “the image of a developing wife”, “the family’s recognition of the wife”, and “support from the neighbors and the local communities”. As the children grew older, the participants became more aware of their role as the father, and as their spouses became increasingly assimilated into Korean society, they felt grateful to them for devoting themselves to their child. In addition, families who felt discomfort and kept their distance from an unfamiliar foreign daughter-in-law before childbirth also recognized the wife as a member of the family through the birth of the child. With these surrounding influences, fathers of multicultural families better recognized their identity and learned their roles.


*When the baby said “Dad” to me for the first time, I was thrilled. After that, the baby used the word “Dad” more than the word “Mom.” When I come home after work, the child runs to me. At that time, I am happy beyond words.*
*(Participant 8)*


*My mother thought a lot of her. My mother thanked her for marrying her old, moneyless son. My mother often said to me, “Because your wife has come to a distant country and suffers, you should love her a lot.”*
*(Participant 10)*


*The relationship between me and my family has improved. All the families gathered on a holiday. At that time, I had my child, so there were many topics for stories. Relatives bought toys for my child, and they loved him. Because I also have a child, meeting with my family is much easier and better than when I had no child.*
*(Participant 2)*


*The creation of a family has made a difference in me. In the past, I didn’t meet people very well. But now, I have met a family similar to ours to eat and talk with. I found a multicultural family similar to ours, and we understand and get more help from such families.*
*(Participant 3)*

### 3.5. Action/Interaction Strategies 

“Accepting the differences and moving forward” were interaction strategies used by the participants in the process of adaptation to fatherhood. Strategies of interaction that emerged as subcategories were “looking back on myself”, “acquiring the role of father”, “acknowledging the differences with the wife”, and “taking a step forward with a family”. As they experience the journey of “finding happiness amid confusion”, they looked back and became aware of their unpreparedness. Although at first, they were unaware of what needs to be done for the child, as fathers, they tried to do their best to meet their financial responsibilities. Apart from working, they tried to participate with their spouses in raising their child. They accepted their differences with their wives and acted as mediators between their wives and mothers who experienced conflict due to linguistic and cultural differences. In order to help spouses adapt, participants spent time with their spouses’ family living in Korea first, so that they could get to know each other. Participants actively met with other multicultural families and tried to participate in events organized by religious or childcare institutions. In doing so, they put their family first, and took these difficult steps while ensuring the happiness of their family.

These interaction strategies appeared in four stages: the “confusion” stage, wherein a man who has married internationally faces difficulty in coming to fully fill the role of a father; the “acceptance of reality” stage, wherein they recognize and strive to learn about their role as a father and the relationship between themselves and their foreign wives; the “adaptation to reality” stage, wherein they recognize their differences and actively helped their wives adapt to life in Korea; and the “growth” stage, wherein they set specific goals and plans for the child’s future and find various ways to establish themselves and their families as members in the Korean society. 


*I regretted choosing a marriage like this at first. But this is my choice. I felt sorry for my wife, because she was also struggling to meet me with no money.*
*(Participant 5)*


*I am working hard to play the role of head of household. When I go to the sea to make money, I have to take a long boat. When I come home after working a long time, I work in the field again. I am living hard. There are two more members of my family who are looking to me, but I am happy to be a father and work for my child.*
*(Participant 1)*


*This baby is my first child, so I try to spend a lot of time with it. But actually, I’m having a hard time. I can’t just leave the child care to my wife, so I try my best to be with her.*
*(Participant 5)*


*After my child was born, I felt more responsible. I don’t have a lot of money, so I have to spend money within limits. I didn’t waste my money before I got married. However, I think my financial responsibilities have increased with the appearance of the baby and the family.*
*(Participant 8)*


*I studied Cambodian language a lot on my cell phone. I can’t write in Cambodian language, but if I use a lot of words in Cambodian, I thought I could talk on the phone. So, we talk on the phone in Cambodian. I’m still studying Cambodian language.*
*(Participant 5)*


*When she had to go to the hospital, I always took her with me. It’s hard to use transportation facilities in the countryside, so I always tried to go with her. It could be dangerous for a pregnant woman as she might fall while riding a bus alone. I thought it was dangerous for the child in that situation. I thought going to the hospital with my wife is the first priority because she can’t speak Korean well.*
*(Participant 1)*

### 3.6. Consequences

As fathers from multicultural families accepted their differences and moved forward, the results followed: “growth with family” and “being made to stay.” The subcategories of “growth with family” are “win-win growth with children” and “win-win growth with spouse,” and the subcategories of “being made to stay” are “crash against the wall of reality” and “live inevitably”.

Participants experienced a variety of emotions such as responsibility and anxiety, as well as feeling indescribable joy on meeting their child for the first time. In addition, they were building their identity as fathers as their children grew up. Participants witnessed their wives trying to quickly adapt to Korean culture during their short honeymoon and nurturing periods. They recognized their wives as their spouses and as the mothers of their children through the latter’s sacrifice for their children. They felt a sense of familial bonding with their spouses. However, they also viewed their situation negatively, worrying about the few work years left to them in occupations exposed to risk.

## 4. Discussion

This qualitative study applied a grounded theory method to develop a substantive theory through which to comprehensively understand and explain the process of paternal adaptation of those who have formed multicultural families. The study results show the problems experienced in this particular cultural context and how various factors relating to the same interact.

In this study, the core category that the Korean men who had formed multicultural families depicted in the process of paternal adjustment were “struggling to protect the family with a double burden”. This refers to a series of processes in which Korean fathers cope with and adapt to the process of fatherhood while experiencing marriage, pregnancy, and the birth and rearing of children. This result is similar to that of previous studies, that men generally experience difficulties while playing the role of a father while raising children but, over time, rediscovered strength and pride, experience satisfaction, and grow [23,24,25]. Therefore, the process of paternal adaptation of fathers of multicultural families is a process of understanding and undertaking the father’s role to lead their family, as in ethnically homogenous families. However, as found in previous studies [6,7], their experience was different from that of the ethnically homogeneous Korean family in that they had to play the universal role of being the father of one family and to play multiple roles required in marrying a foreign woman (head of household, husband, translator, etc.). They not only experienced difficulties in communicating due to linguistic and cultural differences but also felt an additional burden due to their personal backgrounds and difficult economic situations that made it difficult for them to recognize the proper ways of acting as a father.

Another contextual condition, “cultural difference,” includes not only language as an obstacle to effective communication but also reluctance to accompany the wife in the early days of marriage, self-prejudice, and concerns about the appearance of the child. The concerns of these participants included children’s late language development, children experiencing discrimination due to their appearance and bullying among friends, and the social perspective regarding marrying women from underdeveloped countries [12,13,18]. Therefore, multicultural education must move away from language acquisition and cultural experience for migrants toward a direction that includes the cultivation of the multicultural sensibilities of Koreans [26]. In addition, it is necessary to expand economic support for multicultural families and create self-help groups and networks that can increase fathers’ self-esteem. An educational approach that considers cultural differences of multicultural families is required. Cultural roles are important within the social context of both multicultural and native families and influence fatherhood in both contexts [27,28].

In general, fathers experience the feeling of becoming a family and begin to imagine their role as their bond with the fetus begins to grow when they recognize the fetus on the ultrasound image as their own [29]. Most participants in this study were unable to directly participate in prenatal examinations, such as ultrasound imaging, due to geographic conditions or occupational problems, and thus encountered obstacles in the formation of early attachments. For family health in multicultural families, the participation of the husband on behalf of a spouse who has not adapted to the culture and language is essential because pregnancy, childbirth, and postpartum education, which are the initial adjustment periods for the family, have an important impact on family health [12,13]. However, since fathers of multicultural families have serious restrictions in regard to both the time and place to receive this education, an e-learning strategy is needed to accommodate them and to expand the medical services provided to them.

The intervention conditions that hindered active interaction in the process of adaptation to fatherhood in this study were “unprepared for the father’s role”, “child-raising difficulties”, and “family conflict”. The “unprepared for the father’s role” condition made it difficult for the father to acquire a strategy to fulfil his role as a father because it was impossible for him to know how to fulfill the role. This is consistent with the results of previous studies that indicate that not having experienced an ideal caregiver increases the burden felt in the process of adaptation to fatherhood [5,6]. However, even if the father had negative experiences with his own father as a child, a successful paternal adaptation process may provide him with the opportunity to heal [3,7,8]. Therefore, for multicultural families, many of whom have low incomes and lack experience with positive parenting, it is necessary to educate the men in the early stages of marriage before the transition to paternity and mediate through interviews with family counselors.

In parental education for multicultural families, it is necessary to help in actively strengthening attachment behaviors between fathers and children in situations where the role of mothers is limited due to language barriers. The attachment between the child and the father lowers parenting stress arising from the challenge of raising young children [30]. Conflicts between families make it difficult for fathers to play their roles [6,7]. To help fathers develop conflict resolution skills, such as non-violent dialogue, which can enhance parental efficacy without responding in a critical, evasive, or defensive way, related content should be included in parental education programs [31].

In this study, the intervention conditions classified as “powerful” were support from the families and neighbors, as well as community services, which could lead to active adaptation behavior. Previous studies have found that fathers from multicultural families struggle to play their role as fathers more than others due to negative experiences, such as social alienation and prejudice, amid many psychological conflicts and difficulties [6,7]. In particular, previous studies have found that the lack of interest and support from the community causes mental health problems in fathers, decreasing their participation in raising their children [8,9,10]. Therefore, continuous attention from local communities is required, especially for fathers of multicultural families who are exposed to difficulties during this important period [18]. In addition, pregnancy and childbirth are important transitional periods in which parents begin to play a new role in their lives. Hence, treatment, information, and advice are vital for them [11].

However, similar to previous studies [12,13], it was also found that mothers of multicultural families not only have difficulties in acquiring information related to child rearing due to language barriers, but also in accessing medical facilities due to the lack of understanding and discrimination of workers in medical institutions toward them. These difficulties placed a double burden on the father of caring for not only his children, but also his wife. Therefore, there is a need to develop a program to enhance the cultural competence of health care workers [26]. In particular, it is necessary to establish and expand their curriculum to enhance their cultural competence and to provide refresher education for continuous multicultural understanding, because cultural competency of obstetricians, gynecologists, pediatricians, and nurses is required in the process of adaptation to pregnancy, childbirth, and parenting in multicultural families [26].

Fathers from multicultural families used a double strategy of “accepting differences and moving forward” in the process of becoming fathers. This includes efforts to accommodate the wife’s mother tongue as well as the role of the head of the household and participation in parenting activities. Although support policies and programs for married migrant women and their children have been actively provided thus far, the development of educational programs for all family members and for fathers remains insignificant [14]. The father’s active parenting participation reduces the burden on the spouse and prevents problematic behavior in children and facilitates their social skill development [5,19,32]. Therefore, it is necessary to raise social interest in multicultural families through active and practical support for their family health, and develop strategies to improve prenatal, postnatal, and parenting education in the initial adaptation process.

This study has the following limitations. First, the definition of the transitional process, which is constantly evolving with the growth of the children, is limited. This study defined the period of paternity adaptation—the prenatal and one-year postpartum period—as the most important period for family members, during which the process of adaptation took place. Second, it is necessary to be careful about the interpretation of the results since 11 participants do not comprise an adequate sample.

## 5. Conclusions

This is a qualitative study that applied grounded theory to understand the process of paternity adaptation in the case of multicultural families comprising married immigrant women and their Korean husbands, and to present a substantive theory that explains the relationship between a variety of related conditions. It promotes a comprehensive understanding of multicultural families’ negative adaptation, and it lays the foundation for programs and policies to help such families establish themselves as healthy families. This is achieved by understanding the process of Korean fathers’ adaptation to fatherhood and by realizing its interaction with several complex and related conditions, and then developing a substantive theory. This study proposes policies, based on said theory developed through the grounded theory method, for improving medical services in island areas and developing web-based and language-based prenatal, postnatal, and parental education programs, so that multicultural families can establish themselves as healthy families in Korean society. This study can serve as a basis for planning nursing interventions using local resources, while considering the interaction of various factors experienced by Korean fathers in multicultural families during the adaptation to fatherhood.

## Figures and Tables

**Figure 1 ijerph-18-05935-f001:**
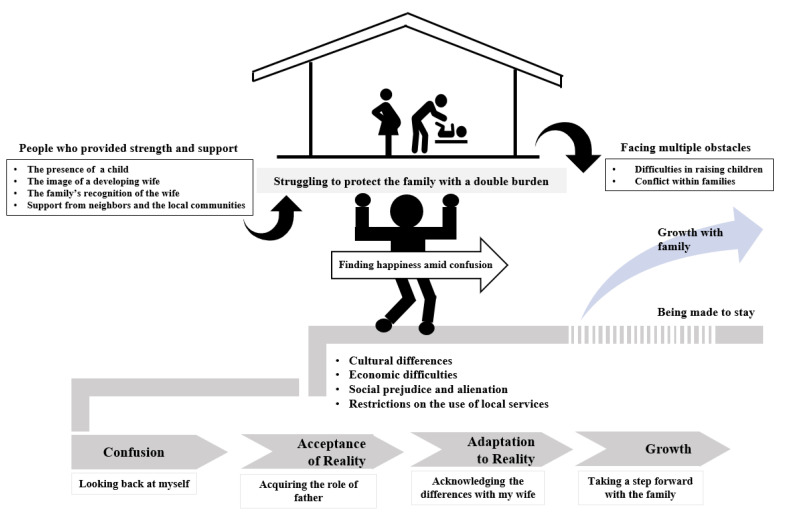
Adaptation process of Korean fathers within multicultural families in Korea.

**Table 1 ijerph-18-05935-t001:** General Characteristics of Participants.

Characteristics	Categories	(N = 11)	
		n (%)	Mean (SD)
Age (in years)	40–49	9 (81.8)	48.09 ± 5.13
	50–59	2 (18.2)	
Age gap between participant and wife (in years)	10–14	4 (36.4)	17.55 ± 4.93
	15–19	4 (36.4)	
	20–25	3 (27.3)	
Occupation	Day laborer	3 (27.3)	
	Public officer	1 (9.1)	
	Driver	5 (45.4)	
	Agriculture and fisheries	2 (18.2)	
Type of family	Nuclear family	11 (100)	
	Large family	0 (0)	
Residential district	Rural area	4 (36.4)	
	Urban area	7 (63.6)	
Wife’s nationality	Vietnamese	5 (45.4)	
	Cambodian	4 (36.4)	
	Filipino	2 (18.2)	
Type of childbirth	Spontaneous delivery	3 (27.3)	
	Cesarean surgery	8 (72.7)	
Type of hospital for childbirth	Women’s hospital	11 (100)	
	Tertiary hospital	0 (0)	
Place of postnatal care	Home	8 (73.7)	
	Hospital + Home	3 (27.3)	
Prenatal education	Participated	2 (18.2)	
	Not participated	9 (81.8)	
Postpartum education	Participated	4 (36.4)	
	Not participated	7 (63.6)	

**Table 2 ijerph-18-05935-t002:** Paradigm, categories, and subcategories of the adaptation process of Korean fathers within multicultural families in Korea.

Subcategories	Categories	Paradigm Element
Continuous suggestions from family and neighbors	Multicultural family reluctantly formed	Causal Conditions
Unwilling marriage		
Being a father by compelling circumstances.		
The confusion of being a father	Finding happiness amid confusion	Central phenomenon
The thrill of being a father		
Different languages	Cultural differences	Contextual conditions
Differences in values		
Differences in nurturing		
Restrictions on economic activities due to aging	Economic difficulties	
The burden of increasing dependents		
Spousal differences in financial views		
Uncomfortable gazes from others at children’s exotic appearance	Social prejudice and alienation	
Negative perceptions of multicultural families		
Medical services in islands with low accessibility	Restrictions on the use of local services	
Services of community institutions that are not easy to use		
Difficulties in raising children	Facing multiple obstacles	Intervention conditions
Conflict within families		
The presence of a child	People who provided strength and support	
The image of a developing wife		
The family’s recognition of the wife		
Support from the neighbors and the local communities		
Looking back on myself	Accepting the differences and moving forward	Action/ Interaction strategies
Acquiring the role of father		
Acknowledging the differences with the wife		
Taking a step forward with the family		
Personal maturity	Growth with family	Consequences
Win-win growth with children		
Win-win growth with spouse		
Crash against the wall of reality	Being made to stay	
Live inevitably		

## Data Availability

The data presented in this study are not publicly available due to privacy concerns.

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
