# Peer review of "Adaptation Process of Korean Fathers within Multicultural Families in Korea"

_ijerph, 2021, doi:10.3390/ijerph18115935_

Round 1

Reviewer 1 Report

Dear authors the topic of this manuscript is interesting. But, 11 fathers are a few sample. How is it possible to generalize the results? I suggest add this in the limits section. Is important for the future to increase the sample.

Best regards

Author Response

Dear Reviewer

We would like to thank you for your time and constructive feedback in improving our manuscript, “Fatherhood Adaptation Process of Korean Fathers with Multicultural

Families in Korea” (ijerph- 1228505), which we submitted to the International Journal of Environmental Research and Public Health. As requested, we include this response to the reviewer’s comments. Please see our responses to the reviewer’s comments after this letter.

We hope that the changes we made have significantly improved the quality of our manuscript. Please do let us know if there is anything else we can do at this stage. We would be more than happy to do everything we can to assist you in this process.

We look forward to hearing from you!

Best Regards,

Hyang-In Cho Chung 

College of Nursing, Chonnam National University

160 Baekseo-ro, Dong-gu, Gwangju 61469, Republic of Korea

[email protected]

Tel: +82-62-530-4956, 010-9262-8759

Fax: +82-62-220-4544

Reviewer 2 Report

Thank you for this novel and very interesting study.

To my opinion, this manuscript is generally well written and contents are treated in-depth. However, to my opinion, redundancies should be avoided as to increase reading fluency. In this regard, a concise style should be prioritized.

Following, my few minor comments:

a) Title: the original title could be slightly changed into: ‘Fatherhood Adaptation Process of Korean Fathers within Multicultural Families in Korea’

b) Abstract - p. 1, line 11: what do you mean with ‘direct observations’? I am not sure this aspect has been addressed well within the main text. P. 1, line 12: ‘..following Strauss and Corbin’. I think ‘Strauss and Corbin’s grounded theory method’ should be specified.

c) Introduction: p. 2, lines 50/1: ‘but also with social perceptions of minorities and weak people in Korean society, where there is a strong sense of being a single people’. To me, this sentence is not clear enough. Please, can you explain better? P. 2, line 91: ‘family, adapt’, comma can be removed.

d) Method: as discussed above, several redundancies can be noticed. Moreover, the subparagraph 2.5 entitled ‘Study validity and researcher preparation’ is very long, and can be appropriately synthesized. I would also suggest to avoid the use of ‘I’, which can be substituted by ‘the first author’.

e) Results: p. 5 lines 212/3: Where can the 86 concepts be retrieved?

f) Conclusion: I think this paragraph could be synthesized, and be mostly focused on practical implications (please, see p. 15, lines 583-594).

Author Response

Dear Reviewer,

We would like to thank you for your time and constructive feedback in improving our manuscript, “Fatherhood Adaptation Process of Korean Fathers with Multicultural

Families in Korea” (ijerph- 1228505), which we submitted to the International Journal of Environmental Research and Public Health. As requested, we include this response to the reviewer’s comments. Please see our responses to the reviewer’s comments after this letter.

We hope that the changes we made have significantly improved the quality of our manuscript. Please do let us know if there is anything else we can do at this stage. We would be more than happy to do everything we can to assist you in this process.

We look forward to hearing from you!

Best Regards,

Hyang-In Cho Chung 

College of Nursing, Chonnam National University

160 Baekseo-ro, Dong-gu, Gwangju 61469, Republic of Korea

[email protected]

Tel: +82-62-530-4956, 010-9262-8759

Fax: +82-62-220-4544

Reviewer 3 Report

First of all, congratulate the authors for carrying out this interesting and necessary study in the socio-community and family sphere. The results are highly enlightening and provide a lot of information to be able to carry out prevention programs in this group, as well as to extrapolate to other groups that are in similar situations.
The work has been developed with great rigor and clarity of exposition.
I would only recommend that the authors review the literature on fatherhood and the influence of cultural roles in both multicultural and native families, in order to carry out a comparison that may benefit multicultural families.

Author Response

(The authors gave the same response as above.)
